# Genome-Wide Identification of DlGRAS Family and Functional Analysis of *DlGRAS10/22* Reveal Their Potential Roles in Embryogenesis and Hormones Responses in *Dimocarpus longan*

**DOI:** 10.3390/ijms262110323

**Published:** 2025-10-23

**Authors:** Guanghui Zhao, Mengjie Tang, Wanlong Wu, Wei Gao, Jinbing Xie, Jialing Wang, Zhongxiong Lai, Yuling Lin, Yukun Chen

**Affiliations:** Institute of Horticultural Biotechnology, Fujian Agriculture and Forestry University, Fuzhou 350002, China; zgh1209507033@163.com (G.Z.); tangmengjie865@163.com (M.T.); wuw1813@163.com (W.W.); gw1398982958@163.com (W.G.); 17816297755@163.com (J.X.); 15106015517@163.com (J.W.); laizx01@163.com (Z.L.)

**Keywords:** *Dimocarpus longan* Lour., GRAS family, hormones responsive, somatic embryogenesis

## Abstract

GRAS family plays a critical role in plant growth and stress responses. In this study, we identified 47 *GRAS* (*DlGRAS*) genes in the longan genome and conducted a comprehensive bioinformatics analysis of these genes. RNA-seq analysis revealed that the expression of these *DlGRAS* genes differed during early SE and across various longan tissues. The quantitative real-time PCR (qRT-PCR) results indicated that the *DlGRAS* genes exhibited differential expression during the early SE of longan, with most of them showing high expression at the globular embryo (GE) stage. Under GA_3_ treatment, the transcript levels of *DlGRAS12/15* decreased significantly. In contrast, exogenous ABA promoted the expression of *DlGRAS6/10*/*23*, indicating that *DlGRAS* genes are responsive to hormones. Compared with CaMV35S-driven *GUS* expression, the promoters of *DlGRAS10/22* increased *GUS* expression, GA_3_ and ABA treatments enhanced promoter activity. DlGRAS10/22 were located in the nucleus. Overexpression of *DlGRAS10/22* in longan SE significantly promoted the transcription levels of SE-related genes, including *DlGID1*, *DlGA20ox2*, *DlLEC1*, *DlFUS3*, *DlABI3* and *DlLEC2*. Therefore, *DlGRAS* may be involved in the early morphogenesis of longan SE through the hormone signaling pathway.

## 1. Introduction

The GRAS family is a plant-specific transcription factor. It plays a crucial regulatory role in various plant biological processes, including plant growth and development [1], stress responses [2], anther microspore formation [3], signal transduction, nodule and mycorrhizal formation, biological detoxification, and other relevant processes [4]. The name derives from the first three members: *GIBBERELLIC-ACID INSENSITIVE* (*GAI*), *REPRESSOR of GAI* (*RGA*) and *SCARECROW* (*SCR*) [5]. The typical GRAS proteins generally consist of 400 to 770 amino acids (aa), comprising a variable N-terminal sequence and a highly conserved C-terminal sequence [6]. The C-terminal 5 motifs (LHRI, VHIID, LHRII, PFYRE, SAW) together form the GRAS domain, with VHIID, PFYRE, and SAW showing higher conservation [7]. The N-terminus of GRAS proteins is a highly variable region that harbors 2 evolutionarily conserved protein domains, DELLA and TVHYNP. Due to the divergence in N-terminal motifs, GRAS proteins can interact with distinct target proteins, thereby contributing to the functional diversity of genes [8]. Members of the GRAS family can be divided into multiple subfamilies according to the structural and functional features of their protein sequences. In *Arabidopsis thaliana*, the GRAS family has been classified into 10 subfamilies: LS, PAT1, SCL3, DELLA, LISCL, SCR, HAM, SHR, SCL4/7 and DLT [9]. In *Secale cereale*, the GRAS family has been divided into 13 subfamilies: LISCL, DLT, OS19, SCL4/7, PAT1, SHR, SCL3, HAM-1, SCR, DELLA, HAM-2, LAS and OS4, with each subfamily exhibiting distinct physiological functions [10].

At present, The GRAS family, having been comprehensively identified and analyzed in model plants, exhibits diverse and conserved physiological functions across species, with its roles validated through functional studies in *Arabidopsis thaliana* [7], *Oryza sativa* [11], *Populus przewalskii* [12], *Eucalyptus grandis* [13], *Cymbidium goeringii* [14], *Solanum lycopersicum* [15], *Secale cereale* [10]. In terms of stress response regulation, GRAS genes enhance plant tolerance to abiotic stresses via multiple mechanisms. In *Betula platyphylla*, *BpPAT1* safeguards the integrity of the cell membrane structure by reducing reactive oxygen species (ROS) accumulation, thereby enhancing salt stress tolerance [16]. Heterologous expression of the *HcSCL13* in *Halostachys caspica* significantly enhances the drought and salt resistance of *Arabidopsis thaliana* [17]. Overexpression of *GmGRAS37* improves the drought and salt tolerance of *Glycine max* [18]. In *Populus davidiana*, *PdbSCL1* interacts with *PdbHAG3*, which catalyses its acetylation, thereby promoting drought tolerance and growth [19]. *GRAS* genes play key roles in biotic stress responses. In *Oryza sativa*, OsGRAS30 acts upstream of OsHDAC1 and interacts with OsHDAC1 to suppress its enzymatic activity. This inhibition increases the histone H3K27ac level, thereby boosting broad-spectrum blast resistance [20]. In hormone signal transduction, *GRAS* genes, especially the DELLA subfamily, are core regulators of gibberellin (GA) signaling. In *Arabidopsis thaliana*, GA is perceived by the nuclear receptor GID1, which then interacts with DELLA proteins and promotes their degradation—this relieves DELLA’s inhibitory effect on downstream transcription factors, regulating plant growth [21]. In *Sorghum bicolor*, the DELLA subfamily participates in GA-mediated growth regulation, while other subfamilies have distinct hormone-related functions [22]. In *Arabidopsis thaliana*, *AtLAS* regulates axillary meristem formation and controls GA metabolism by modulating the expression of *GA2ox4* [23,24]. In growth and development, *GRAS* genes govern critical processes such as meristem maintenance, organ formation, and embryonic development. SCR interacts with SHR to regulate root radial growth [25], while HAM maintains the undifferentiated state of apical meristem stem cells to support continuous differentiation [26]. In *Populus trichocarpa*, *PtrDLT* specifically regulates xylem differentiation [27]. For crop yield-related traits, *ZmGRAS11* controls kernel development: loss-of-function mutants show delayed cell expansion and reduced kernel size/weight, while overexpression lines exhibit the opposite phenotype [28]. In *Solanum lycopersicum*, silencing the *SlGRAS4* gene under high-temperature stress reduces heat tolerance [29], and down-regulating *SlGRAS26* alters plant architecture and flowering time [30], demonstrating the gene’s role in balancing stress response and reproductive development. Notably, *GRAS* genes have shown regulatory potential in embryonic development. For example, in *Lilium brownii*, the GRAS protein LlSCL participates in the transcriptional regulation of anther microspore development [3], while in *A. thaliana*, DELLA proteins interact with the key embryonic regulator LEC1 (LEAFY COTYLEDON1) during late embryogenesis: GA-mediated DELLA degradation relieves its inhibitory effect on LEC1, thereby promoting embryonic development [31].

Longan (*Dimocarpus longan* Lour.) belongs to the *Dimocarpus* of Sapindaceae, primarily distributed in tropical and subtropical regions. For longan research, the somatic embryogenesis (SE) system serves as an effective solution to overcome the challenges of difficult sampling and inconsistent material quality of embryos under natural conditions—critical limitations that have hindered studies on its embryonic development. Notably, the status of embryonic development directly impacts longan fruit yield and quality, making this process biologically and agronomically significant [32,33,34]. Previously, Chen [35] reported that *DlGAI* and *DlGAIP-B*, 2 members of the DELLA subfamily, not only regulate longan seed germination and floral organ development but also participate in hormone transduction and stress response mechanisms, highlighting the functional diversity of *GRAS* genes in longan. While existing research on *GRAS* genes has largely focused on their roles in growth, development, and stress responses in model plants (*Arabidopsis thaliana*, *Solanum lycopersicum*, *Populus davidiana*, *Zea mays* and *Oryza sativa*), their involvement in longan SE remains underexplored. To address this gap, we conducted a comprehensive bioinformatics analysis of the DlGRAS family using longan genome and transcriptome data, with the goal of elucidating the physicochemical properties, expression patterns, and biological functions of its members. Additionally, we performed subcellular localization verification and promoter function analysis to further clarify the potential functions of *DlGRAS* genes. Collectively, these efforts lay a foundation for deciphering the regulatory mechanisms underlying the DlGRAS family’s role in longan SE.

## 2. Results

### 2.1. Diverse Physical and Chemical Properties of DlGRAS Family Members

Utilizing TBtools software (V2.031) and NCBI Blastp, potential DlGRAS family members were identified from the third-generation longan genome database. Through multiple alignment of amino acid sequences and verification of DlGRAS-related domains via NCBI Blastp and Conserved Domains Database (CDD), 47 valid DlGRAS members were finally confirmed. Based on their chromosome location information, these 47 *DlGRAS* genes were named *DlGRAS1~DlGRAS47* (Appendix A).

Physicochemical properties of DlGRAS family indicated that none of the DlGRAS members possessed a signal peptide or a transmembrane domain, indicating that they were probably not a secreted protein (Appendix A). The molecular weight of the protein ranged from 24281.44~86620.45 D, with the number of amino acids varying between 212~761 aa. The isoelectric point fell within the range of 4.62 to 6.70. The instability coefficient ranged from 35.65 to 60.44. The hydrophilicity coefficient of DlGRAS ranged from −0.595 to −0.05, with the exception of DlGRAS22, which had a hydrophilicity coefficient of 0.057, classifying it as a hydrophobic protein. Subcellular localization predictions indicated that DlGRAS were predominantly localized in the nucleus.

### 2.2. High Homology of DlGRAS Exists in Litchi and Longan

To further explore the evolutionary relationship of the *DlGRAS* family, the phylogenetic tree of DlGRAS, including longan (47) and *Arabidopsis thaliana* (32), was constructed by MEGA5.0 software (Figure 1A). Based on the evolutionary relationship, the *DlGRAS* family were classified into 10 subfamilies: LISCL, SCL3, DELLA, HAM, SCR, PAT1, SCL4/7, DLT, SHR and LAS. Among these subfamilies, the LISCL subfamily comprised 11 members, while the DLT, LAS and the SCL4/7 subfamilies each contained only 1 member. Given the distinct distribution patterns of DlGRAS members within the clustering relationships, it is plausible to speculate that these patterns are associated with the diverse biological functions of each subfamily.

The collinearity analysis was conducted both within longan species and between litchi and longan species to further elucidate their evolutionary relationships. The results demonstrated that *DlGRAS* genes were unevenly distributed on 12 chromosomes within the longan species, of which 25 genes were distributed on DlChr5, DlChr7 and DlChr12, accounting for approximately 53.2% of the total genes (Figure 1B). A total of 14 members within the *DlGRAS* family were identified to participate in 10 tandem duplication events. These duplication pairs are as follows: *DlGRAS3* and *DlGRAS8*, *DlGRAS28* and *DlGRAS37*, *DlGRAS28* and *DlGRAS5*, *DlGRAS30* and *DlGRAS41*, *DlGRAS36* and *DlGRAS40*, *DlGRAS30* and *DlGRAS7*, *DlGRAS37* and *DlGRAS5*, *DlGRAS41* and *DlGRAS7*, *DlGRAS46* and *DlGRAS11*, *DlGRAS23* and *DlGRAS27.* The collinearity analysis between longan and litchi revealed that *DlGRAS* genes in litchi were also distributed on 12 chromosomes. Moreover, 38 pairs of genes exhibited significant collinearity with *DlGRAS*, indicating a close evolutionary relationship between longan and litchi (Figure 1C).

### 2.3. DlGRAS Exhibits Remarkable Conservation During Evolutionary Processes

The results of the analyses focusing on conserved motifs, protein conserved domains and gene structure of the *DlGRAS* are presented in Figure 2. Gene structure analysis revealed striking variability in intron number, with 72.3% of *DlGRAS* genes being intronless, a finding consistent with the results of *GRAS* studies in other plants [36,37]. This high intronless rate suggests that *DlGRAS* genes may have evolved via horizontal gene transfer from prokaryotes or undergone intron loss during evolution. Notably, the number of introns varies among different subfamilies of DlGRAS, and this subfamily-specific gene structure may reflect functional divergence. Following the prediction of conserved motifs in the DlGRAS proteins, it was observed that motif 3 was distributed among 47 members, indicating that motif 3 is highly conserved within the DlGRAS family. In conjunction with the aforementioned analyses, it was revealed that members belonging to the same subfamily of *DlGRAS* exhibited similar gene structures. This similarity in gene structure implies that the functions of each subfamily within *DlGRAS* are distinct. In summary, the evolution of *DlGRAS* balances the strong conservation of core motifs/domains and the subfamily-specific divergence of gene structures, enabling longan to adapt to unique developmental and environmental requirements.

### 2.4. Multiple Hormones and Transcription Factors Regulate DlGRAS Expression

The binding sites of transcription factors for the *DlGRAS* were predicted, and their interactions with transcription factors were further scrutinized. A total of 17 transcription factors were identified (Figure 3A). Notably, there were variations in the number, type, and distribution of transcription factors in longan promoters of *DlGRAS*. Specifically, the binding sites for BBR-BPC, MIKC-MADS, and Dof transcription factor were extensively distributed in the promoter regions of *DlGRAS*. In contrast, the Nin-like transcription factor binding site was exclusively found in the promoter region of *DlGRAS23*. Based on the analysis results mentioned above, it can be inferred that the promoter regions of *DlGRAS* had transcription factor binding sites and played a vital role in regulating gene expression.

To further understand the functional disparities among the promoters of *DlGRAS*, the PlantCARE online tool was used to analyze the *cis*-elements in the 2000 bp promoter sequence upstream of the initiation codon of the DlGRAS family in longan. The results demonstrated that the promoter regions of *DlGRAS* contained multiple response elements involved in plant growth and development, hormone responses, stress tolerance, and light signaling (Figure 3B). Specifically, 98% of *DlGRAS* possessed light-responsive elements, indicating *DlGRAS* can respond to light signals, thus influence the growth and development of longan. Furthermore, the vast majority of members exhibited responsiveness to hormone regulation. Approximately 68% of the members responded to abscisic acid, while 50% responded to gibberellin. These findings suggest that the *DlGRAS* genes may participate in diverse hormone signal transduction pathways and have distinct roles in hormone response.

### 2.5. DlGRAS Transcription Factor Coordinates SE, Organogenesis and Stress Adaptation in Longan

Distinct expression patterns were observed among different genes of the *DlGRAS* at various stages of early SE (Figure 4A). *DlGRAS6/15/28* were continuously upregulated from EC to GE; *DlGRAS10* was specifically expressed only at the ICpEC stage. *DlGRAS22/26/27* were downregulated from EC to GE and 10 genes were highly expressed in NEC but low in SE stages. Based on these unique expression profiles associated with SE progression, 9 core candidate genes (*DlGRAS6*, *DlGRAS10*, *DlGRAS12*, *DlGRAS15*, *DlGRAS22*, *DlGRAS23*, *DlGRAS26*, *DlGRAS27*, *DlGRAS28*) were finally screened, implying their differential regulatory roles in distinct SE stages. *DlGRAS* genes exhibited varying expression patterns in different tissues (Figure 4B). *DlGRAS2/19/45* were highly expressed in roots, *DlGRAS3* in leaves, *DlGRAS32/47* specifically in pericarp, and *DlGRAS44* in seeds. *DlGRAS22* was highly expressed in flower buds and *DlGRAS26* in flowers, while all *DlGRAS* genes showed low expression in stems. These results indicated that candidate *DlGRAS* genes might participate in specific organogenesis processes.

12 *DlGRAS* genes were highly expressed in darkness. *DlGRAS3/9/31* were upregulated under blue light, and *DlGRAS4/5/12* were upregulated under white light, indicating that these genes might respond to light signals to regulate SE (Figure 4C). Under different temperature treatments, most of the *DlGRAS* genes were highly expressed at 35 °C (Figure 4D). It is speculated that high temperature can up-regulate the expression of *DlGRAS* genes to counteract or repair the damage caused by stress, thereby ensuring the normal growth and development of longan.

### 2.6. Differential Regulation of Longan Embryo Development by DlGRAS Genes

Based on the results of RNA-seq analysis, it was observed that, in comparison to other genes, these 9 genes (*DlGRAS6*, *DlGRAS10*, *DlGRAS12*, *DlGRAS15*, *DlGRAS22*, *DlGRAS23*, *DlGRAS26*, *DlGRAS27* and *DlGRAS28*) potentially play a vital role in the process of longan SE. Consequently, we conducted qRT-PCR assays on these 9 genes to further validate their expression patterns during longan embryogenesis. The relative expression levels of *DlGRAS6*, *DlGRA26* and *DlGRAS27* exhibited a gradual increase from EC to GE stage, indicating that *DlGRAS* play a positive regulatory role during longan SE (Figure 5B). In contrast, the relative expression levels of *DlGRAS10* and *DlGRAS23* decreased rapidly from EC to GE stage and then increased, with the highest expression level observed at the GE stage. Specifically, the specific high expression of *DlGRAS22* at the EC stage indicates its crucial role in the induction and maintenance of longan EC. *DlGRAS12*, *DlGRAS15* and *DlGRAS28* displayed the same expression trend. In summary, the *DlGRAS* may play different roles during the early SE of longan.

The qRT-PCR analysis of *DlGRAS* at different developmental stages of zygotic embryos revealed that *DlGRAS* could be expressed from the S1 to S8 stages and exhibited diverse expression patterns (Figure 5D). The expression profiles of *DlGRAS6* and *DlGRAS26* exhibited distinct stage-specific patterns during zygotic embryo development. Specifically, their expression levels were elevated at the S5 stage and markedly decreased at the S8 stage. Notably, the expression level of *DlGRAS10* demonstrated a significant up-regulation throughout the S2 to S8 period of zygotic embryo development compared to that at the S1 stage. Regarding the expression patterns of *DlGRAS12*, *DlGRAS23*, *DlGRAS27* and *DlGRAS28*, they displayed a high degree of similarity. All 4 genes were highly expressed at the S7 stage, indicating a potential coordinated regulatory mechanism at this particular developmental stage. During the transition from S1 to S4, *DlGRAS15* and *DlGRAS22* exhibited an identical expression trend. However, a divergence in their expression patterns was observed from S4 to S8. Based on these observed expression patterns, it can be inferred that *DlGRAS* gene may play crucial regulatory roles at different developmental stages of zygotic embryos, potentially contributing to the precise control of embryo development processes.

### 2.7. DlGRAS Integrates GA3 and ABA Signaling to Regulate Longan SE

Previous studies have indicated that *DlGRAS* genes may be implicated in hormone signaling pathways. Therefore, we examined the expression levels of *DlGRAS* genes in longan EC treated with GA_3_ and ABA. This was done with the aim of elucidating the specific regulatory pathways involving *DlGRAS* genes. The qRT-PCR analysis of longan EC treated with varying concentrations of GA_3_ are presented in Figure 6A. As the concentration of GA_3_ increased, the transcription levels of *DlGRAS6*, *DlGRAS10*, *DlGRAS23*, *DlGRAS26*, *DlGRAS27* and *DlGRAS28* exhibited a significant upregulation, indicating that GA_3_ can promote the expression of the these genes. The transcriptional level of *DlGRAS12* was completely suppressed, while the expression of *DlGRAS22* was inhibited only under the treatment of 9 mg/L GA_3_. The qRT-PCR analysis of *DlGRAS* under ABA treatment showed that the relative expression levels of *DlGRAS6*, *DlGRAS10*, *DlGRAS23* and *DlGRAS26* upregulated after treatment with 3~12 mg/L ABA. However, within this concentration range, the transcription level of these genes generally showed a decreasing trend with the increase in ABA concentration (Figure 6B). The relative expression levels of *DlGRAS12*, *DlGRAS15*, *DlGRAS27* and *DlGRAS28* were significantly lower than those of the CK, indicating that ABA treatment inhibited their transcription levels. *DlGRAS22* significantly promoted its transcription level only when treated with 1 mg/L ABA. Hence, *DlGRAS* genes can respond to GA_3_ and ABA hormones.

### 2.8. DlGRAS10 and DlGRAS22 Play an Important Role in the Nucleus

Our findings revealed that *DlGRAS10* and *DlGRAS22* exhibited distinct expression patterns. Specifically, when compared with other genes, the expression level of *DlGRAS10* was notably higher at the S2~S8 stages of zygotic embryo development than that at the S1 stage. Meanwhile, the expression level of *DlGRAS22* at the EC stage, which represents the early phase of SE, was significantly elevated compared to the other two developmental stages under investigation. Based on these observations, we hypothesize that *DlGRAS10* and *DlGRAS22* may assume more crucial roles than other genes in the relevant biological processes. To gain a deeper understanding of the potential biological functions of the *DlGRAS10* and *DlGRAS22* genes, we conducted an investigation into the subcellular localization of their corresponding proteins in the inner epidermis of onion (Figure 7). In pCAMBIA1302:GFP, the fluorescence signal was uniformly distributed throughout the onion inner epidermis. In contrast, in the pCAMBIA1302-DlGRAS10:GFP and pCAMBIA1302-DlGRAS22:GFP, the fluorescence signal was exclusively detected in the nucleus, indicating DlGRAS10 and DlGRAS22 play a crucial role within the nucleus.

### 2.9. Exogenous GA3 and ABA Modulate DlGRAS10 and DlGRAS22 Promoter Activities During Longan SE

The core promoter regions and transcription start site of *DlGRAS10* and *DlGRAS22* were predicted by the BDGP (https://fruitfly.org/seq_tools/promoter.html). The prediction results showed that both *DlGRAS10* and *DlGRAS22* possess type A transcription start sites. Specifically, *DlGRAS10* has two transcription start sites, labeled as A and C, while *DlGRAS22* has two types of transcription start sites, namely A and T (Appendix A).

The pCAMBIA1301-DlGRAS10pro::*GUS* and pCAMBIA1301-DlGRAS22pro::*GUS* vectors were constructed and transiently transformed into *N. benthamiana* by *Agrobacterium*-mediated transformation (Figure 8B). A blue color was observed in *N. benthamiana* leaves in the GUS staining assay, indicating the expression of the *GUS* gene in these leaves (Figure 8A). The results demonstrated that both *DlGRAS10* and *DlGRAS22* promoters were capable of activating and enhancing the expression of downstream *GUS* genes. Notably, *DlGRAS10* had a more significant driving ability for *GUS* gene expression (Figure 8C). The transiently transformed *N. benthamiana* were sprayed with different hormones, with water serving as the control. The results indicated that *DlGRAS10* and *DlGRAS22* promoted the expression of downstream *GUS* genes in response to ABA and GA_3_ signals. Among the treatments, both *DlGRAS10* and *DlGRAS22* showed the most significant ability to drive *GUS* gene expression under ABA treatment. Under ABA treatment, the driving ability of *DlGRAS10* for *GUS* gene expression was approximately 4 times that of the control group, while the ability of *DlGRAS22* to activate *GUS* gene expression was about 3 times that of the control group. Therefore, it can be inferred that *DlGRAS* promoters may affect the SE of longan by responding to hormones and enhancing the expression of downstream *GUS* gene.

### 2.10. DlGRAS10/22 Transient Expression Activates Embryogenic Transcriptional Networks and Modulates Endogenous ABA/GA3 Homeostasis

To further elucidate the mechanisms by which *DlGRAS10* and *DlGRAS22* function in SE, we conducted transient expression of *DlGRAS10* and *DlGRAS22*. Subsequently, we examined their regulatory relationships with other genes known to be involved in SE. GUS staining revealed that *DlGRAS10*-OE and *DlGRAS22*-OE exhibited a distinct blue coloration, whereas the control group did not appear color (Figure 9A). PCR-based molecular identification of *DlGRAS10/22*-OE lines confirmed the amplification of specific bands corresponding to the *Hyg* and *GUS* genes within the range of 300~500 bp (Figure 9B), indicating successful integration of the recombinant plasmid into *Agrobacterium tumefaciens* and infection of the longan EC. The qRT-PCR showed that the *GUS* gene was highly expressed in both *DlGRAS10*-OE and *DlGRAS22*-OE lines. Specifically, the expression level of the *GUS* gene in *DlGRAS10*-OE was 1.86 times higher than that in the control group, while in *DlGRAS22*-OE, it was approximately 20 times higher (Figure 9D). *DlGRAS10* and *DlGRAS22* genes were significantly upregulated in their respective overexpressing lines. The expression level of *DlGRAS10* in *DlGRAS10*-OE was about 116 times higher than that in the control group, whereas the expression level of *DlGRAS22* in *DlGRAS22*-OE was about 23 times higher.

To investigate the regulatory relationship between *DlGRAS10/22* and other SE-related genes, the expression profiles of *DlGID1*, *DlGA20ox2*, *DlLEC1*, *DlFUS3*, *DlABI3* and *DlLEC2* were analyzed in transgenic longan EC. The results indicated a significant enhancement in the transcription levels of these genes (Figure 9D). Specifically, the expression levels of *DlGID1*, *DlGA20ox2*, *DlFUS3, DlABI3* and *DlLEC2* in *DlGRAS10*-OE were about 7.5 times, 2.1 times, 2 times, 2.2 times and 2.1 times higher, respectively, compared to the control group. In *DlGRAS22*-OE, the expression level of *DlLEC1* was the highest, reaching 3.12 times that of the control group, while the expression level of *DlABI3* was about 3.4 times higher. These findings suggest that *DlGRAS10/22* may affect longan SE through reciprocal regulation with SE-related genes, including *DlGID1*, *DlGA20ox2*, *DlLEC1*, *DlFUS3*, *DlABI3* and *DlLEC2*.

In order to study whether *DlGRAS10* and *DlGRAS22* genes are involved in hormone biosynthesis pathways, the contents of relevant hormones were measured in *DlGRAS10*-OE and *DlGRAS22*-OE cell lines. The contents of ABA and GA_3_ were determined (Figure 9C). The results showed that the contents of ABA and GA_3_ increased in *DlGRAS10/22*-OE lines. Therefore, *DlGRAS10/22* may collectively influence the process of longan SE by coordinately regulating the metabolism and signaling pathways of ABA and GA_3_.

## 3. Discussion

### 3.1. DlGRAS Family May Exhibit Evolutionary Conservation

The GRAS family of transcription factors is paramount importance in plant growth, development, hormone signaling, and stress responses. Elucidating the biological functions of GRAS genes in longan holds significant scientific value. In this study, a total of 47 DlGRAS family members were identified through bioinformatics approaches. The phylogenetic tree constructed for the DlGRAS family clustered its members into 10 distinct branches. It is postulated that these branches are associated with the diverse biological functions of different subfamilies. For instance, scarecrow (SCR) and short-root (SHR) have been demonstrated to co-regulate endothelial development and stem cell maintenance [6,38]. The LlSCL protein is implicated in the transcriptional regulation of microspore development in *Lilium browni* anthers [3]. The LAS subfamily is known to regulate the development of axillary bud meristems in plants [6], while the HAM subfamily can maintain the growth of adventitious buds in flowering plants [39]. During the course of plant evolution, it has been hypothesized that genes lacking introns may exhibit greater evolutionary conservation [40,41]. Previous studies have revealed that a substantial proportion of GRAS genes in various plant species are intron-less. For example, *Arabidopsis thaliana* [12], *Gossypium hirsutum* [42], *Secale cereale* [10], *Avena sativa* [43] and *Castanea mollissima* [44], most GRAS genes lack introns. In the present study, only 13 DlGRAS genes contained introns, resulting in an intron-less rate of 72.3%. This finding is consistent with the intron-less characteristics observed in the GRAS gene of *A. thaliana*, indicating that the GRAS genes may have evolved from prokaryotes via horizontal gene transfer and gene replication events [45,46]. Studies have indicated that RGL1, RGL2, RGL3, RGA1 and GAI within the AtGRAS family belong to the DELLA subfamily [6]. The DELLA protein acts as an inhibitory factor in the GA signaling pathway. Endogenous GA relies on the DELLA domain within the DELLA protein to induce its degradation, thereby relieving its inhibitory effect [47,48]. The N-terminus of DlGRAS1, DlGRAS23 and DlGRAS27 contain a DELLA domain. Given that these genes belong to the DELLA subfamily, they may participate in GA signal transduction during the growth and development of longan. Tandem repeats and segmental repeats are recognized as the primary mechanisms driving gene family expansion. Genes are retained in plant genomes through these mechanisms play an essential roles in development and environmental stress responses [49,50]. The collinearity analysis revealed a total of 10 tandem repeat events within the DlGRAS in longan. Through the prediction and experimental verification of subcellular localization of DlGRAS protein sequences, it was determined that these proteins are predominantly located in the nucleus. This localization pattern suggests that DlGRAS function within the nucleus, and their functions are likely to be evolutionarily conserved.

### 3.2. DlGRAS Genes May Be Involved in Longan SE Through Hormone Responses

Gene expression patterns can offer vital information for predicting gene functions [51]. The results of qRT-PCR demonstrated that *DlGRAS* exhibited different expression trends under GA_3_ treatment, indicating its potential significant role in SE via the gibberellin signal transduction pathway. Previous studies have indicated that when plants are subjected to environmental stresses, the expression of corresponding stress-responsive genes is activated, and ABA, a vital plant stress-signaling hormone, also accumulates in plants [52]. The *cis*-elements in the promoter region play an important role in gene regulation and expression [53]. To adapt to changes in the external environment and respond to stress responses, the combination of transcription factors and *cis*-elements can regulate the expression of related responsive genes, thereby improving plant resistance to stresses [54]. The promoter *cis*-elements of *DlGRAS* were predicted, revealing that *DlGRAS* possessed light-responsive elements as well as various hormone-responsive elements, such as those for auxin, gibberellin, abscisic acid, and salicylic acid. Through the construction and functional analysis of *DlGRAS* vector, it was found that compared with *GUS* expression driven by the CaMV35S, the promoters of *DlGRAS10/22* increased *GUS* expression. Moreover, GA_3_ and ABA treatments enhanced promoter activity (Figure 10). In conclusion, the *DlGRAS* genes may be involved in the process of longan SE through hormone responses.

### 3.3. DlGRAS10/22 Affected Longan SE by Regulating SE-Related Genes and Hormone Synthesis

During SE, the *GRAS* gene is regulated by hormone signals and environmental factors, and interacts with other transcription factors; these regulatory and interactive effects collectively modulate embryonic cell differentiation, proliferation, and development. The GA signaling pathway regulates signals through the formation of the GA-GID1-DELLA complex and DELLA degradation [55]. Additionally, the DELLA protein interacts with GAF1 to co-regulate the target gene *GA20ox2* [56]. The GA signal suppressor, DELLA, can directly interact with the key regulator *LEAFY COTYLEDON1* (*LEC1*) during late embryonic development. GA-mediated *DELLA* degradation relieves its inhibitory effect on *LEC1*, thereby promoting embryonic development [31]. In *Arabidopsis*, *AtLEC2* can activate the expression of *ABI3* and *FUS3* [57]. The *ABI3* gene can regulate ABA synthesis, and *FUS3* also modulates the dynamic balance of ABA/GA in plants by either inhibiting GA synthesis or promoting ABA accumulation [58]. In this study, overexpression of *DlGRAS10/22* significantly enhanced the transcription levels of *DlGID1*, *DlGA20ox2*, *DlLEC1*, *DlFUS3*, *DlABI3* and *DlLEC2*. Moreover, the contents of GA_3_ and ABA increased in *DlGRAS10/22*-OE lines (Figure 10). It is hypothesized that the interaction between *DlGRAS* gene and other SE-related genes affects the longan SE by signal transduction pathway.

**Figure 10 ijms-26-10323-f010:**
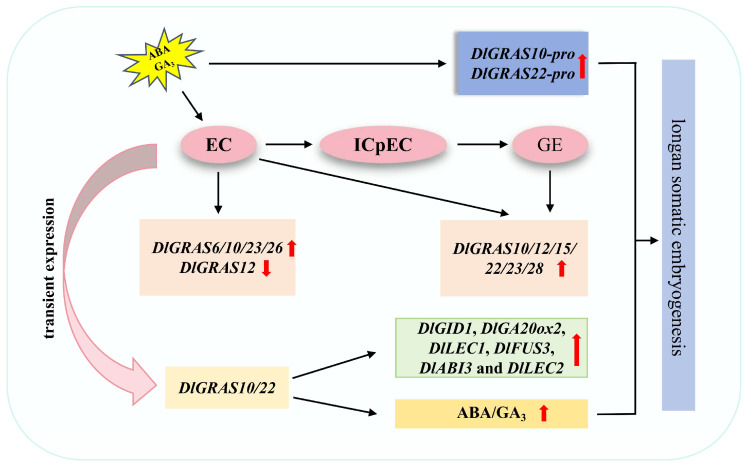
The molecular mechanism of *DlGRAS* gene regulating longan SE. The red arrow indicates that the gene expression is up-regulated or down-regulated under this treatment, and the endogenous hormone level is increased.

## 4. Materials and Methods

### 4.1. Materials

Embryogenic callus (EC), incomplete compact pro-embryogenic cultures (ICpEC), and globular embryos (GE) of longan were successfully obtained by following the protocol established by Lai et al. [32]. In accordance with the method described by Tang et al. [59], different developmental stages of zygotic embryos were collected. Additionally, longan EC was subjected to treatments with varying concentrations of GA_3_ and ABA. The third-generation of longan genome data were downloaded from NCBI (BioProject accession: PRJNA792504; genome sequence of *D. longan*: SRR17675476) [60]. Furthermore, the genome data of *Arabidopsis thaliana* were retrieved from TAIR (http://arabidopsis.org) (9 November 2022) [61], and the genome data of *Litchi chinensis* [62] were downloaded from (https://data.mendeley.com/datasets/kggzfwpdr9/1) (15 November 2022).

### 4.2. Identification of DlGRAS in Longan

The GRAS protein domain model (PF03514) was obtained from the Pfam database (http://pfam-legacy.xfam.org/) (2 December 2022) [63]. Subsequently, the HMMER3.0 software was utilized to construct hidden Markov model for the GRAS protein. Utilizing an E-value threshold of less than 10^−5^ as the criterion, potential members of the DlGRAS family were identified and sifted from the third-generation sequences of the longan genome database. The GRAS protein sequences of *A. thaliana* were selected as the query sequence. Blastp homology comparison was performed on the aligned protein sequences through the NCBI and the Conserved Domains Database (CDD, https://www.ncbi.nlm.nih.gov/cdd) (20 December 2022) [64] was utilized for protein domain prediction, and members lacking DlGRAS-related domains were excluded from further analysis. The TBtools software was used to visualize [65].

The molecular weight (MW), number of amino acids (aa), isoelectric point (PI), instability index (Instability index), and average hydrophilicity (GRAVY) of DlGRAS were predicted by ExPASy software (https://web.expasy.org/protparam) [66]. The SignalP (http://www.cbs.dtu.dk/services/SignalP/) (2 January 2023) [67] was applied to identify the signal peptide of DlGRAS proteins. The subcellular localization of the DlGRAS protein sequences was predicted by WOLF PSORT (https://wolfpsort.hgc.jp/) (10 January 2023) [68]. Additionally, the transmembrane domain (TM) of DlGRAS protein was predicted by TMHMM Server2.0 (http://www.cbs.dtu.dk/services/TMHMM/) (22 January 2023) [69]. Chromosome localization analysis and visualization of the *DlGRAS* were performed using TBtools.

### 4.3. Phylogenetic Relationship and Collinearity Analysis of DlGRAS in Longan

To construct the phylogenetic tree of the GRAS family between longan and *Arabidopsis*, the neighbor-joining (NJ) method in MEGA5.0 software was used. The analysis settings included 1000 bootstrap replicates, a Poisson model, and partial deletion with 95% cutoff. The resulting phylogenetic tree was refined and visualized using the online tool iTOL (https://itol.embl.de/upload.cgi) (25 January 2023) [70]. TBtools software was used to visualize both the intraspecific collinearity within longan and the interspecific collinearity between longan and litchi.

### 4.4. Analysis of Conserved Motifs, Domains and Gene Structure of DlGRAS

The conserved motifs of DlGRAS protein were analyzed using the online MEME software (https://memesuite.org/meme/tools/meme) (27 January 2023) [71], with the number of motifs set to 10 and other parameters set to their default values. The gene structure of *DlGRAS* was analyzed using the GSDS online software (https://gsds.gao-lab.org/) (6 February 2023) [72]. Protein domain prediction was carried out using the CDD. The results were visualized using TBtools software.

### 4.5. Transcription Factor Binding Site and Cis-Element Analysis of DlGRAS

The 2000 bp upstream sequence from the ATG start codon of 47 *DlGRAS* genes was extracted using TBtools software, and this sequence was designated as the promoter sequence. The *cis*-elements within these promoter sequences were analyzed using the online software PlantCARE (http://bioinformatics.psb.ugent.be/webtools/plantcare/html/) (10 February 2023) [73]. Additionally, the PlantTFDB software (http://planttfdb.cbi.pku.edu.cn) (17 February 2023) [74] was used to predict transcription factors that potentially regulate the target genes, threshold *p*-value ≤ 1 × 10^−4^. TBtools software was utilized for the visualization of the results.

### 4.6. Expression Analysis of DlGRAS During Early SE and Different Tissues/Light Qualities/Abiotic Stresses

In accordance with the method described by Tang et al. [59], the FPKM values of *DlGRAS* genes were obtained for different stages during early SE (NEC [non-embryogenic callus], EC [embryogenic callus], ICpEC [incomplete compact pro-embryogenic cultures], GE [globular embryos]) (NCBI BioProject number: PRJNA891444) [75]. Additionally, FPKM values were collected for various tissues (NCBI BioProject number: PRJNA326792) (root, stem, flower bud, flower, leaf, young fruit, peel, seed), different light qualities [76] (blue, white, black), different temperatures (15 °C, 25 °C, 35 °C) (NCBI BioProject number PRJNA889670). The FPKM values were normalized using the log_2_FPKM transformation. The heatmap was generated to visualize the expression patterns using TBtools software.

### 4.7. RNA Extraction and qRT-PCR Analysis of DlGRAS

Total RNA was extracted from samples during the early stage of longan SE and under different hormone treatments using the TransZolUp kit (TransGenBiotech, Beijing, China). First-strand cDNA was synthesized using the Hifair^®^ III 1st Strand cDNA Synthesis SuperMix for qPCR (gDNA digester plus) (Yeasen, Shanghai, China). Primers were designed using Primer3 software (https://primer3.ut.ee/cgi-bin/primer3/primer3web_results.cgi) (13 February 2023) [77] and DNAMAN 6.0 software (Appendix A). *DlACTB*, *DlEF-la,* and *DlUBQ* were selected as internal reference genes. The qRT-PCR reaction system had a total volume of 20 μL, consisting of 8.2 μL ddH_2_O, 10 μL HRbioTM qPCR SYBR^®^Green Master Mix (No Rox) (Heruibio, Guangzhou, China), 0.4 μL specific primer pairs, and 1 μL 10-fold diluted cDNA. The operating parameters of the qRT-PCR were as follows: 95 °C for 30 s, followed by 40 cycles of 95 °C for 10 s, and 58 °C for 30 s. Each reaction was performed with 3 biological replicates. The data obtained from qRT-PCR were analyzed, and the relative expression levels of each gene were calculated using the 2^−ΔCT^ method [78].

### 4.8. Subcellular Localization of DlGRAS10/12

The subcellular localization of DlGRAS protein sequences was predicted by WOLF PSORT (https://wolfpsort.hgc.jp/). The CDS sequences of *DlGRAS10* and *DlGRAS22* were subjected to restriction endonuclease analysis. Primers were designed utilizing DNAMAN 6.0 software (Appendix A). The pCAMBIA1302-GFP vector was constructed following the method described by Zhang et al. [79]. The recombinant plasmid carrying the target gene was transferred into *Agrobacterium*. Subsequently, *DlGRAS10* and *DlGRAS22* were transiently expressed in *Allium cepa* epidermal cells via *Agrobacterium*-mediated approach. The samples were incubated in the dark environment at 28 °C for 3 d. The inner epidermis of the onion was meticulously dissected using blades and tweezers, and immersed in 4’,6-diamidino-2-phenylindol (DAPI) solution for 5 min, and then soaked in distilled water for 1 min. The fluorescence signal localization of DlGRAS10 and DlGRAS22 was observed using Olympus FV1200 confocal laser microscope (Olympus, Tokyo, Japan).

### 4.9. Functional Analysis of DlGRAS10 and DlGRAS22 Promoters

The core promoter regions and transcription start sites of *DlGRAS10* and *DlGRAS22* were predicted using the BDGP (https://fruitfly.org/seq_tools/promoter.html) (19 June 2023) [80]. Primers were designed with DNAMAN 6.0 software, encompassing the core promoter region (Appendix A). The pCAMBIA1301-DlGRAS10pro::*GUS* and pCAMBIA1301-DlGRAS22pro::*GUS* vectors were constructed and transiently expressed in *Nicotiana benthamiana* via the *Agrobacterium*-mediated method. The plants were cultured in the dark for 24 h and then exposed to light for another 24 h. Exogenous hormones, namely 50 μM ABA and GA_3_, were evenly sprayed on the abaxial surface of transgenic *N*. *benthamiana* leaves, with water serving as the control. Samples were collected after 24 h of light culture. Subsequently, GUS staining and relative expression detection were carried out.

### 4.10. DlGRAS10/22 Transiently Transformed to Longan EC

Primers for the CDS sequences of *DlGRAS10*/*22* were designed using the DNAMAN online tool and the pCAMBIA1301-*GUS* vector was constructed (Appendix A). The plasmids of pCAMBIA1301-DlGRAS10-*GUS* and pCAMBIA1301-DlGRAS22-*GUS* were separately transformed into *Agrobacterium tumefaciens* (EHA105). A single colony was selected for bacterial liquid PCR verification, and the positive clone bacterial liquid was expanded and cultured. The bacteria were collected by centrifugation at 5000 r/min for 10 min. The cells were resuspended in a MS (Coolaber, Beijing, China) suspension containing 3% sucrose, 200 mM acetosyringone (AS), 100 mM MgCl_2_ and the OD_600_ was adjusted to 0.6~0.8. The longan EC with favorable growth after 15 d of culture was selected and transferred to the prepared bacterial solution for infection for 30 min. The bacterial solution was filtered using filter paper and allowed to dry. After the longan EC had dried, it was transferred to MS solid medium containing 100 mM AS for 3 d. After this period, GUS staining was performed, and the results were observed using fluorescence microscope. The common longan EC was used as a control. The transiently transformed longan EC was sampled and stored at −80 °C.

### 4.11. RNA, DNA Extraction and Molecular Identification of Transiently Transformed Longan EC

Total RNA from the transiently transformed longan EC was extracted using the BioTeke kit (BioTeke, Wuxi, China) and reverse transcribed into cDNA using Revertaid Master Mix (Thermo Fisher Scientific, Shanghai, China). DNA was extracted using FastPure Plant DNA Isolation Mini Kit (Vazyme, Nanjing, China), and DNA was used as template for resistance gene detection.

### 4.12. Expression Analysis of Transiently Transformed Longan EC

Primers were designed as described above (Appendix A), *DlACTB* was used as the internal reference gene, and the relative expression level was calculated using the 2^−ΔΔCT^ method. Significance analysis was performed using SPSS, and the graphs were generated using GraphPad Prism 8.0.2 software.

### 4.13. Determination of Endogenous Hormones in Transiently Transformed Longan EC

0.1 g sample was accurately weighed into a mortar, and 1 mL of phosphate-buffered saline (PBS) with pH = 7.0 was added for grinding. The contents of GA_3_ and ABA were determined using an Elisa kit (Shanghai Enzyme Linked Technology Co., Ltd., Shanghai, China). The specific operational procedures method and calculation formula are detailed in the respective kit instructions.

### 4.14. The Statistical Analysis

In this study, all data were analyzed via one-way analysis of variance (ANOVA) using SPSS 20 software, followed by post hoc comparisons. GraphPad Prism 8.0.2 software was employed for data visualization and figure generation. In the figures, distinct lowercase letters denote statistically significant differences among treatment groups, with a significance level set at *p* < 0.05.

## 5. Conclusions

In this study, a total of 47 *DlGRAS* were identified based on *D. longan* genome data. RNA-seq and qRT-PCR analyses demonstrated that *DlGRAS* genes exhibited distinct expression patterns during early SE of longan, with specific responses to exogenous GA_3_ and ABA. Subcellular localization confirmed that DlGRAS10 and DlGRAS22 localized to the nucleus. Promoter functional assays showed that the promoters of *DlGRAS10/22* drove higher *GUS* expression than the CaMV35S promoter, and this activity was significantly enhanced by GA_3_ and ABA treatments. Transient overexpression of *DlGRAS10/22* in longan EC increased endogenous GA_3_ and ABA contents, and significantly upregulated the transcription of SE-related genes (*DlGID1*, *DlGA20ox2*, *DlLEC1*, *DlFUS3*, *DlABI3*, *DlLEC2*). Collectively, these findings indicate that *DlGRAS* genes, particularly *DlGRAS10* and *DlGRAS22*, participate in regulating early SE of longan by mediating hormone (GA_3_/ABA) signaling pathways and modulating the expression of SE-related genes.

## Figures and Tables

**Figure 1 ijms-26-10323-f001:**
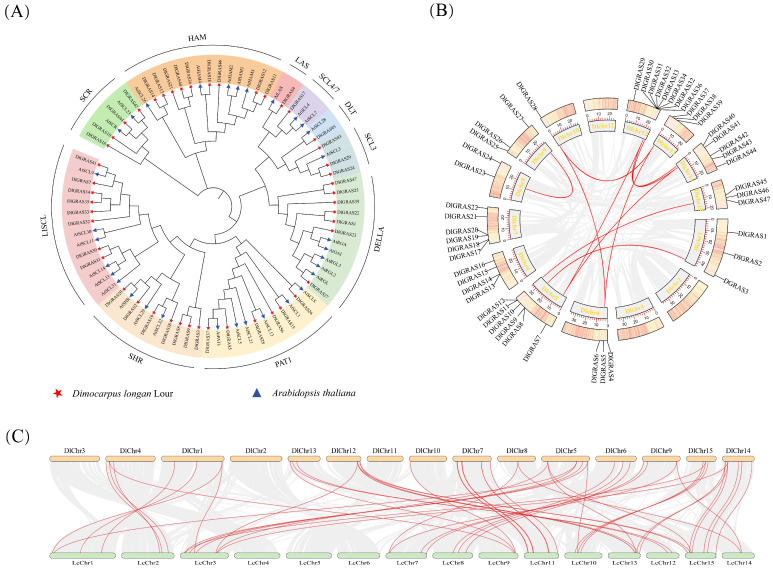
Phylogenetic tree and collinearity analysis of *DlGRAS*. (**A**) The phylogenetic tree of DlGRAS proteins in longan (Dl) and *A. thaliana* (At). (**B**) The collinearity analysis of *DlGRAS* genes in longan. (**C**) The collinearity analysis of *DlGRAS* between longan (Dl) and litchi (Lc). The genes connected by red arcs exhibit collinearity, and the gray arc regions represent all the collinearity regions in the longan genome.

**Figure 2 ijms-26-10323-f002:**
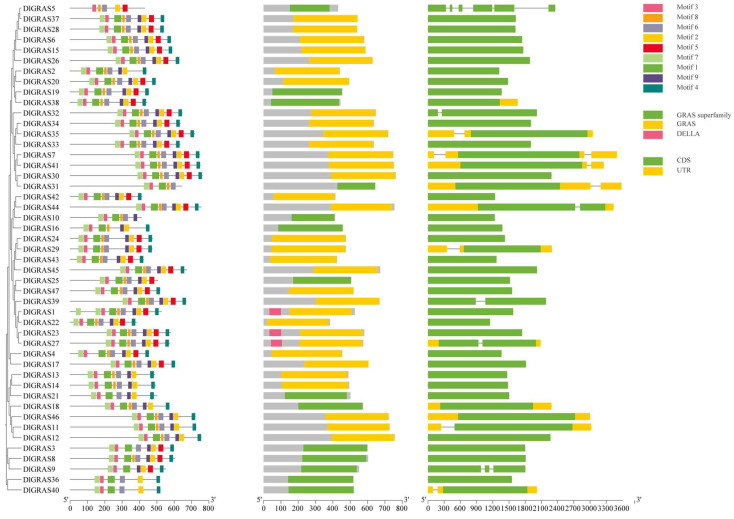
Protein conserved motifs, protein conserved domains and gene structure of *DlGRAS*. The motifs numbered from 1~9 are displayed as boxes in different colors; different colors represent different domains; the green boxes represent exons; the yellow boxes represent UTR; the black lines represent introns.

**Figure 3 ijms-26-10323-f003:**
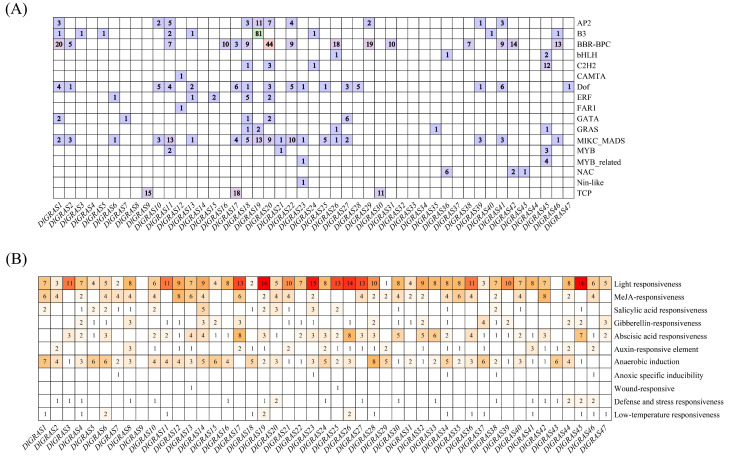
Transcription factor binding sites and *cis*-elements of *DlGRAS*. (**A**) The transcription factor binding site of *DlGRAS* genes. (**B**) The *cis*-elements of *DlGRAS*. The blank spaces represent the absence of elements, the color intensity represents the number of components, and the number in the square represents the number of elements.

**Figure 4 ijms-26-10323-f004:**
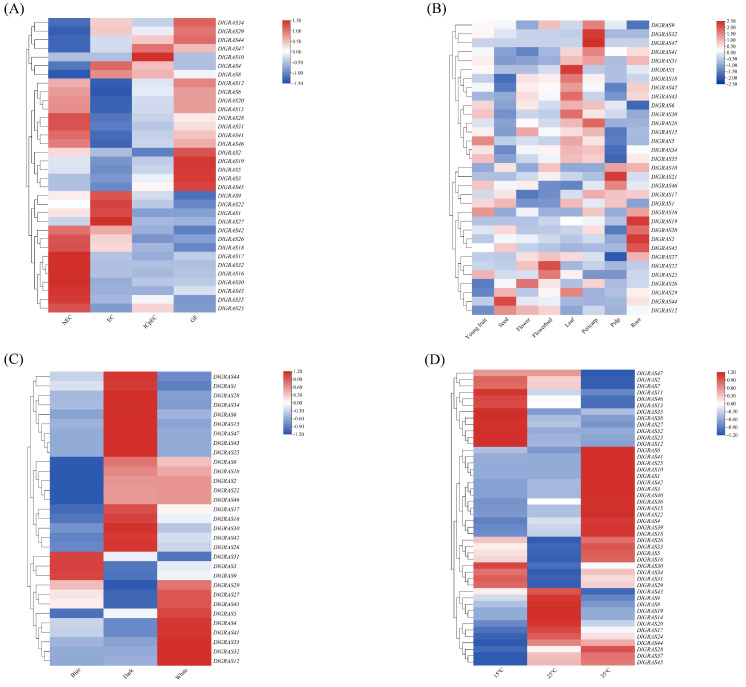
Expression patterns of *DlGRAS* genes based on FPKM values. (**A**) The specific expression of *DlGRAS* genes during different SE stages. (**B**) The particular expression of *DlGRAS* in other tissues. (**C**) The specific expression of *DlGRAS* under different light qualities. (**D**) The particular expression of *DlGRAS* at various temperatures. All plant materials were ‘Honghezi’ longan. Different colors on the scale bar are log_2_FPKM, which represent different transcript levels.

**Figure 5 ijms-26-10323-f005:**
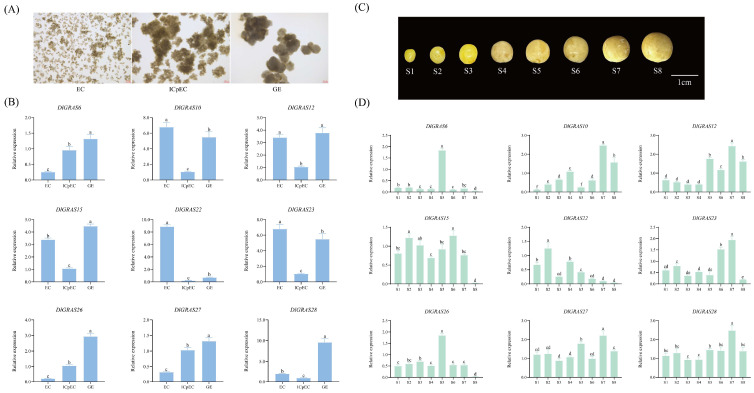
The qRT-PCR analysis of *DlGRAS* during early SE and different developmental stages of zygotic embryos. (**A**) Morphology during the early stage of SE in longan. The scale bar represents 200 μm. (**B**) The qRT-PCR analysis of *DlGRAS* during the early SE. (**C**) Morphology at different developmental stages of zygotic embryos. The scale bar represents 1 cm. (**D**) The qRT-PCR analysis of *DlGRAS* at different developmental stages of zygotic embryos. The internal reference genes were *DlACTB*, *DlEF-la*, *DlUBQ*, three biological replicates, EC: embryogenic callus; ICpEC: Incomplete embryogenic compact structure; GE: globular embryo, different lowercase letters (a, b, c, d, e) show significant differences between treatments (*p* < 0.05).

**Figure 6 ijms-26-10323-f006:**
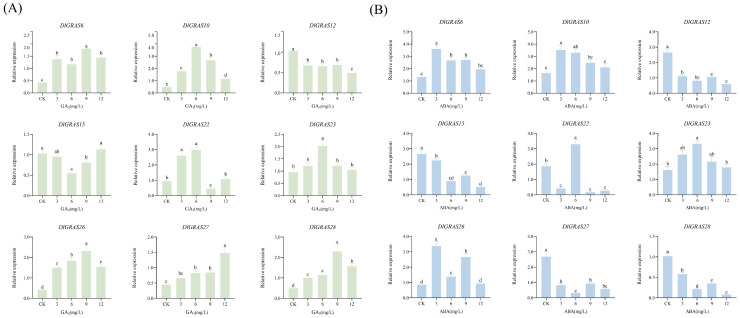
The expression of *DlGRAS* under different exogenous hormone treatments. (**A**) The qRT-PCR analysis of *DlGRAS* under GA_3_ treatment. (**B**) The qRT-PCR analysis of *DlGRAS* under ABA treatment. The internal reference genes were *DlACTB*, *DlEF-la*, and *DlUBQ*, three biological replicates, different lowercase letters (a, b, c, d, e) show significant differences between treatments (*p* < 0.05).

**Figure 7 ijms-26-10323-f007:**
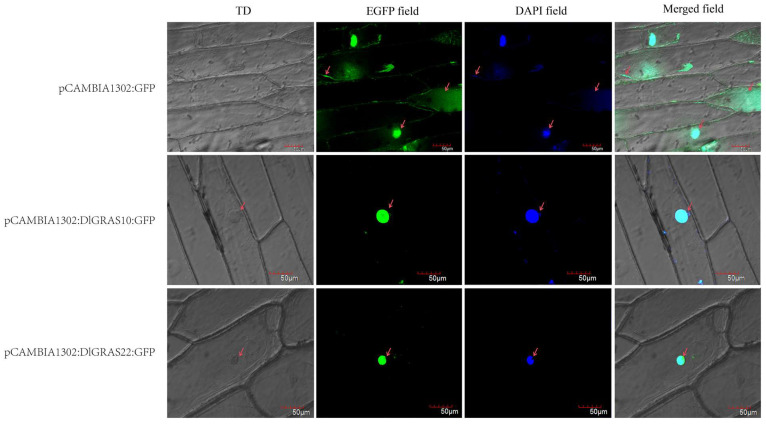
Subcellular localization of DlGRAS proteins in onion epidermal cells. TD represents the transmission light channel. The scale bar is 50 μm, and the arrows indicate the localization of GFP and DAPI fluorescence signals within the cells. The excitation wavelength for GFP is 475 nm, and for DAPI, it is 450 nm.

**Figure 8 ijms-26-10323-f008:**
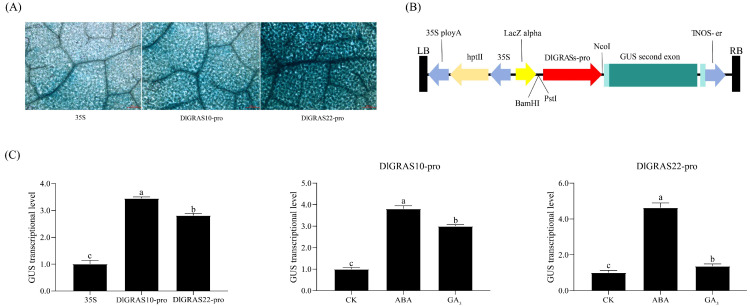
*GUS* gene expression level of *DlGRAS* under ABA and GA_3_ treatments. (**A**) The histochemical staining the *GUS* gene in *N. benthamiana* leaves, the scale bar represents 200 μm. (**B**) The schematic diagram of the *DlGRAS* promoter vector. (**C**) The expression level of the *GUS* gene driven by 35S and *DlGRAS* under ABA and GA_3_ treatments. The internal reference gene was *Nb18S*, three biological replicates, and different lowercase letters (a, b, c) show significant differences between treatments (*p* < 0.05).

**Figure 9 ijms-26-10323-f009:**
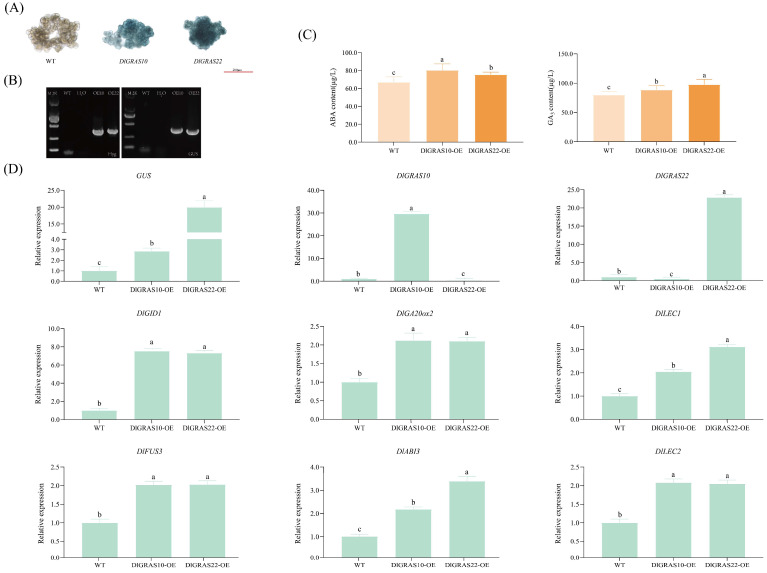
*DlGRAS10/22* Transformation of longan EC. (**A**) GUS histochemical staining, the scale bar is 200 μm. (**B**) DNA molecular identification of transiently transformed longan EC, M2K represents the length of Maker is 2000 bp. (**C**) Determination of ABA and GA_3_. (**D**) The expression of *GUS* and SE-related genes in transgenic longan EC. The internal reference gene was *DlACTB*, three biological replicates, and different lowercase letters (a, b, c) show significant differences between treatments (*p* < 0.05).

## Data Availability

All data generated or analyzed during this study are included in this published article.

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
