# Peer review of "Genome-Wide Identification of DlGRAS Family and Functional Analysis of DlGRAS10/22 Reveal Their Potential Roles in Embryogenesis and Hormones Responses in Dimocarpus longan"

_ijms, 2025, doi:10.3390/ijms262110323_

Round 1
Reviewer 1 Report
Comments and Suggestions for Authors
Overall, the authors have conducted a thorough analysis of the DlGRAS gene family in longan. The methods are appropriate and the figures are well presented. The manuscript is recommended for acceptance after minor revisions. Specific revision suggestions are as follows:
1. Reference 30: It is difficult to understand the relationship between the reference 30 and the sample acquisition in this MS. The authors are advised to carefully check it.
2. 2.2 section: What is the accession ID of the GRAS protein domain model? The authors are advised to list it.
3. 2.3 section: "1000 repeated tests" should be changed to "1000 bootstrap replicates".
4. 2.6 section: How was the FPKM value obtained? A simple method description or a clear reference to previous studies should be provided. The "2" in "log2FPKM" should be a subscript, and "FPKM" should be in normal font rather than a superscript.
5. There is a separate section (2.14) for statistical analysis methods. There is no need to list the statistical test methods again in section 2.7.
6. 2.7 section: The qPCR reaction conditions should be described in detail.
7. 2.7 and 2.12 sections: Why were both 2-ΔCT and 2-ΔΔCT methods used? Please explain.
8. 2.8 section: "A. cepa" appears for the first time. The full name of the genus should be provided.
9. 3.1 section: Dlo_022103.1, Dlo_007688.1, Dlo_025233.1, dlo_037243.1, dlo_039202.1, Dlo_001945.1 and Dlo_001946.1. Are "Dlo" and "dlo" the same?
10. 3.7 section: Why does the change in the expression of DIGRAS genes upon GA3 and ABA treatment of SE indicate that "DlGRAS genes may play a key regulatory role in longan SE through the gibberellin or ABA signal transduction pathway"? At this stage, the evidence presented cannot support this conclusion. In my opinion, the change in the expression of DlGRAS genes in this treatment is influenced by both SE development and GA3 or ABA treatment. The authors can conclude that DlGRAS genes can respond to GA3 or ABA hormones.
Author Response
Dear Reviewer:
Thank you very much for your comments. Those comments are all insightful and very helpful for revising and improving our manuscript, as well as the important guiding significance to our research. We have studied comments carefully and have revised the manuscript thoroughly, and the point-by-point responses to the comments are as follows.
All changes to the revised manuscript are indicated in the manuscript using track changes.
Reference 30: It is difficult to understand the relationship between the reference 30 and the sample acquisition in this MS. The authors are advised to carefully check it.
Author’s response: Thank you so much for your comments. We have revised in the manuscript. The ‘Hong He Zi’ (‘HHZ’) longan ECs, which involved the embryogenic callus (EC), incomplete compact pro-embryogenic cultures (ICpECs), and globular embryo (GE) were obtained as previously described by Lai et al.
2.2 section: What is the accession ID of the GRAS protein domain model? The authors are advised to list it.
Author’s response: Thank you for your comments. We have added this in the revised manuscript. (Line 439)
2.3 section: "1000 repeated tests" should be changed to "1000 bootstrap replicates".
Author’s response: Thank you for your comments. We have rewrite this sentence in the revised manuscript. (Line 460)
2.6 section: How was the FPKM value obtained? A simple method description or a clear reference to previous studies should be provided. The "2" in "log2FPKM" should be a subscript, and "FPKM" should be in normal font rather than a superscript.
Author’s response: Thank you for your comments. We have added this in the revised manuscript. In accordance with the method described by Tang et al., Excel software (12.1.0.16120) was used to extract the FPKM values of DlGRAS family members from the transcriptomes of early SE (NEC, EC, ICpEC, and GE) (NCBI BioProject number: PRJNA891444), different tissues (NCBI BioProject number: PRJNA326792) (young fruit, seed, flower, flower bud, leaf, pulp, root, and stem), different light qualities (blue, white, dark as the control), and different temperatures (15 â—¦C, 25 â—¦C, and 35 â—¦C) (NCBI BioProject number: PRJNA889670), normalized by log2FPKM. And then, We have revised in the manuscript. (Line 480-483)
There is a separate section (2.14) for statistical analysis methods. There is no need to list the statistical test methods again in section 2.7.
Author’s response: Thank you for your comments. We have revised this in the revised manuscript.
2.7 section: The qPCR reaction conditions should be described in detail.
Author’s response: Thank you for your comments. We have added this in the revised manuscript. (Line 498-499)
2.7 and 2.12 sections: Why were both 2-ΔCT and 2-ΔΔCT methods used? Please explain.
Author’s response: Thank you for your comments. 2-ΔCT method in Section 2.7: It calculates the relative expression level of target genes normalized to internal reference genes (DlACTB, DlEF-1α, DlUBQ) in a single sample. 2-ΔΔCT method in Section 2.12: It measures the fold change of target gene expression between a treatment group (transiently transformed longan EC) and a control group.
2.8 section: "A. cepa" appears for the first time. The full name of the genus should be provided.
Author’s response: Thank you so much for your careful check. We have revised this in the revised manuscript. (Line 508)
3.1 section: Dlo_022103.1, Dlo_007688.1, Dlo_025233.1, dlo_037243.1, dlo_039202.1, Dlo_001945.1 and Dlo_001946.1. Are "Dlo" and "dlo" the same?
Author’s response: Thank you for your comments. "Dlo" and "dlo" are the same. In accordance with the suggestions provided by other reviewers, this section of content has been removed.
3.7 section: Why does the change in the expression of DIGRAS genes upon GA3 and ABA treatment of SE indicate that "DlGRAS genes may play a key regulatory role in longan SE through the gibberellin or ABA signal transduction pathway"? At this stage, the evidence presented cannot support this conclusion. In my opinion, the change in the expression of DlGRAS genes in this treatment is influenced by both SE development and GA3 or ABA treatment. The authors can conclude that DlGRAS genes can respond to GA3 or ABA hormones.
Author’s response: Thank you for your rigorous suggestions. We have revised those in the manuscript. (Line 267)
Furthermore, all changes to the revised manuscript are indicated in the manuscript using track changes. Thank you so much.

Reviewer 2 Report
Comments and Suggestions for Authors
The paper systematically elucidated and identified changes in the GRAS gene of longan in embryogenesis and hormones responses. The research is academically valuable and innovative, demonstrating substantial effort, detailed data and reliable conclusions. However, the manuscript requires considerable refinement in terms of writing quality, figure presentation and attention to detail. The current version detracts from the reading experience and hinders comprehension of the excellent research content. Comprehensive and meticulous revision before acceptance were recommended. Specific suggestions are listed below to assist the authors in refining the manuscript:
- The second paragraph of the introduction section should not simply present an unstructured compilation of previous research findings. Instead, it should systematically address the diverse functions of GRAS genes and how they have been validated in different crops. For example, it should explain the roles that GRAS genes play in stress responses and in which crops they have been demonstrated. Where relevant, it should also highlight key advances in hormone response and embryonic development studies in other plant species.
- Gene names should be italicized, while proteins should be in Roman type. There are numerous gene name formatting issues in the introduction section; please review the entire text meticulously for the correct use of italics and Roman type.
- Latin species nomenclature usage: In the third paragraph of the introduction, the family name should not be italicized. Instead, italicize the genus (wrong in Dimocarpus of Sapindaceae ). The first occurrence of the Latin name (A. cepa) in Section 2.8 requires the full scientific name. Authors should carefully review all Latin names throughout the manuscript and amend them according to guidelines.
- In the 'Materials and Methods' section (2.1) and the 'Results' section (3.1), the authors performed genome-wide identification on both second- and third-generation genomes. However, the most accurate final results were obtained from the third-generation genome, yielding 47 genes. It is therefore recommended that references to the second-generation genome are omitted and it is stated directly that the precise results were derived from the third-generation genome.
- All bioinformatics tools, websites and software used in the 'Materials and Methods' section require citation.
- The calculation method for relative expression (2-ΔCT) in Section 2.7 of the Materials and Methods is incorrectly stated and should be consistent with Section 2.12.
- There are multiple instances of inappropriate conjunction usage and grammatical phrasing that. Authors should also refine their sentences and correct errors. Examples include: 1) The improper use of 'conversely' in the second paragraph of the introduction;2) Section 2.14 should use the passive voice for data analysis ('all data were analyzed') rather than using 'data' as the subject;3) Section 3.4 requires 'above' to be used as an adverb at the end of the sentence.
- There are multiple formatting issues in the titles and main text that require adjustment. The title of Section 3.2 is embedded within the body text, and inconsistent font sizes appear in several instances. These require careful review and correction.
- Figure citations and captions are incorrectly formatted. The abbreviation ‘fig.’ in ‘Fig. 1a’ requires a space. While the figures use lowercase letters, the captions use uppercase letters. Consistency must be maintained.
- Figure 3a depicts transcription factor binding sites, and Figure 3b shows cis-acting elements. However, both figures include descriptive statistics for the MYB element. Which set of results is reliable? Could this redundant presentation be avoided? It is recommended that transcription factors do not appear in the cis-element diagram.
- Section 3.5 should be condensed to focus on how RNA-seq data were used to screen 9 candidate genes and analyze their expression patterns across tissues, light conditions and temperatures. Excessive repetition of transcriptomic data under various conditions obscures the core message.
- Similarly, while the first sentence of Section 3.6 states that 9 genes were screened, Section 3.5 lacks any description of the screening process or how these genes were identified.
- Why does Section 3.9 only cite Fig. 8c, with no corresponding description for Figs 8a and 8b?
- The discussion in Section 4.1 refers to ten tandem repeat events, yet Section 3.1 contains no description of replication patterns. Please supplement this information in the Results section.
- The conclusions section should state the key findings directly and draw conclusions without excessive elaboration on the experimental procedures.

The quality of English must be improved.
Author Response
Dear Reviewer:
Thank you very much for your comments. Those comments are all insightful and very helpful for revising and improving our manuscript, as well as the important guiding significance to our research. We have studied comments carefully and have revised the manuscript thoroughly, and the point-by-point responses to the comments are as follows.
All changes to the revised manuscript are indicated in the manuscript using track changes.
- The second paragraph of the introduction section should not simply present an unstructured compilation of previous research findings. Instead, it should systematically address the diverse functions of GRAS genes and how they have been validated in different crops. For example, it should explain the roles that GRAS genes play in stress responses and in which crops they have been demonstrated. Where relevant, it should also highlight key advances in hormone response and embryonic development studies in other plant species.
Author’s response: Thank you for your rigorous suggestions. We have revised those in the manuscript. (Line 49-84)
- Gene names should be italicized, while proteins should be in Roman type. There are numerous gene name formatting issues in the introduction section; please review the entire text meticulously for the correct use of italics and Roman type.
Author’s response: Thank you for your rigorous suggestions. We have revised those in the manuscript.
- Latin species nomenclature usage: In the third paragraph of the introduction, the family name should not be italicized. Instead, italicize the genus (wrong in Dimocarpus of Sapindaceae ). The first occurrence of the Latin name (A. cepa) in Section 2.8 requires the full scientific name. Authors should carefully review all Latin names throughout the manuscript and amend them according to guidelines.
Author’s response: Thank you so much for your comments. We have revised those in the manuscript. (Line 85, 508)
- In the 'Materials and Methods' section (2.1) and the 'Results' section (3.1), the authors performed genome-wide identification on both second- and third-generation genomes. However, the most accurate final results were obtained from the third-generation genome, yielding 47 genes. It is therefore recommended that references to the second-generation genome are omitted and it is stated directly that the precise results were derived from the third-generation genome.
Author’s response: Thank you for your rigorous comment. We have revised those in the manuscript. (Line 107-112)
- All bioinformatics tools, websites and software used in the 'Materials and Methods' section require citation.
Author’s response: Thank you for your comments. We have added this in the revised manuscript.
- The calculation method for relative expression (2-ΔCT) in Section 2.7 of the Materials and Methods is incorrectly stated and should be consistent with Section 2.12.
Author’s response: Thank you for your comments. 2-ΔCT method in Section 2.7: It calculates the relative expression level of target genes normalized to internal reference genes (DlACTB, DlEF-1α, DlUBQ) in a single sample. 2-ΔΔCT method in Section 2.12: It measures the fold change of target gene expression between a treatment group (transiently transformed longan EC) and a control group.
- There are multiple instances of inappropriate conjunction usage and grammatical phrasing that. Authors should also refine their sentences and correct errors. Examples include: 1) The improper use of 'conversely' in the second paragraph of the introduction;2) Section 2.14 should use the passive voice for data analysis ('all data were analyzed') rather than using 'data' as the subject;3) Section 3.4 requires 'above' to be used as an adverb at the end of the sentence.
Author’s response: Thank you so much for your comments. We have revised in the manuscript. (Line 173, 556)
- There are multiple formatting issues in the titles and main text that require adjustment. The title of Section 3.2 is embedded within the body text, and inconsistent font sizes appear in several instances. These require careful review and correction.
Author’s response: Thank you so much for your careful check. We have carefully reviewed and revised the entire manuscript.
- Figure citations and captions are incorrectly formatted. The abbreviation ‘fig.’ in ‘Fig. 1a’ requires a space. While the figures use lowercase letters, the captions use uppercase letters. Consistency must be maintained.
Author’s response: Thank you for your valuable and insightful comment. We have revised in the manuscript.
- Figure 3a depicts transcription factor binding sites, and Figure 3b shows cis-acting elements. However, both figures include descriptive statistics for the MYB element. Which set of results is reliable? Could this redundant presentation be avoided? It is recommended that transcription factors do not appear in the cis-element diagram.
Author’s response: Thank you for your rigorous comment. Both sets of results are reliable. To avoid such redundancy, we have deleted the overlapping parts between the cis-element map and transcription factor binding sites. (Figure 3)
- Section 3.5 should be condensed to focus on how RNA-seq data were used to screen 9 candidate genes and analyze their expression patterns across tissues, light conditions and temperatures. Excessive repetition of transcriptomic data under various conditions obscures the core message.
Author’s response: We gratefully appreciate for your valuable suggestion. We have revised those in the manuscript. (Line 193-209)
- Similarly, while the first sentence of Section 3.6 states that 9 genes were screened, Section 3.5 lacks any description of the screening process or how these genes were identified.
Author’s response: Thank you for your comments. We have added this in the revised manuscript. (Line 193-199)
- Why does Section 3.9 only cite Fig. 8c, with no corresponding description for Figs 8a and 8b?
Author’s response: Thank you so much for your careful check. We have added this in the revised manuscript. (Line 301-302)
- The discussion in Section 4.1 refers to ten tandem repeat events, yet Section 3.1 contains no description of replication patterns. Please supplement this information in the Results section.
Author’s response: We gratefully appreciate for your valuable suggestion. We have added those in the manuscript. (Line 134-138)
- The conclusions section should state the key findings directly and draw conclusions without excessive elaboration on the experimental procedures.
Author’s response: Thank you for your valuable and insightful comment. We have revised n the manuscript. (Line 562-572)
Furthermore, all changes to the revised manuscript are indicated in the manuscript using track changes. Thank you so much.
Reviewer 3 Report
Comments and Suggestions for Authors
This manuscript (ijms-3875083) presents experimental results in exploring regulatory genes in genome of Dimocarpus longan Lour., which belong to GRAS family, mostly highlighting studies of the promoter function, gene expression, and subcellular localization of few DlGRAS regulators. The experiments were designed well and conducted accordingly. Surely, the results presented are novel, informative and very interesting in expanding our understanding of the regulatory functions of GRAS-family regulators in plant development and growth, especially works on differential regulation of longan embryo development by 9 DlGRAS genes and DlGRAS proteins’ sub-localization (DlGRAS 10 and DlGRAS22). However, the improvement could still be made in presenting the experimental results, mostly highlighting and comparatively to the previous works. It can be considered for acceptance for publication after minor reversion.
A couple of comments for consideration:
- Presenting results in comparison DlGRAS transcription factors to GRAS family regulators from other model plants, be more specific and clearer. Not only addressing the evolutional conservation (most highly conserved domains, motif and elements etc.), it needs to highlight the specificity in sequence variation and functions of certain DlGRAS genes, especially from different sub-families. For example, the sub-title “GRAS genes play a pivotal regulatory role during early-stage SE and organogenesis”, readers would be confusing whether this is for review paper and presenting new results.
- Need to define “DlGRAS’ when firstly mentioning in the Introduction or even in Abstract. Also be consistent with DlGRAS or GRAS (more general) in text.
- The reviewer doesn’t know why Yin et al and Zhou et al were highlighted in citation of works in Introduction. Surely, there must be a reason for scientific significance or work mostly related to the current research project. No doubt, it’s fine if their works were first reported, not reviewing other’s works. Just making sure. Also, in updating previous discoveries of GRAS family transcriptional factors, more considerations should be focused on the original works, even from reviews.
Author Response
Dear Reviewer:
Thank you very much for your comments. Those comments are all insightful and very helpful for revising and improving our manuscript, as well as the important guiding significance to our research. We have studied comments carefully and have revised the manuscript thoroughly, and the point-by-point responses to the comments are as follows.
All changes to the revised manuscript are indicated in the manuscript using track changes.
- Presenting results in comparison DlGRAS transcription factors to GRAS family regulators from other model plants, be more specific and clearer. Not only addressing the evolutional conservation (most highly conserved domains, motif and elements etc.), it needs to highlight the specificity in sequence variation and functions of certain DlGRAS genes, especially from different sub-families. For example, the sub-title “GRAS genes play a pivotal regulatory role during early-stage SE and organogenesis”, readers would be confusing whether this is for review paper and presenting new results.
Author’s response: We gratefully appreciate for your valuable suggestion. We have revised those in the manuscript. (Line 37-48, 149-154, 159-161)
- Need to define “DlGRAS’ when firstly mentioning in the Introduction or even in Abstract. Also be consistent with DlGRAS or GRAS (more general) in text.
Author’s response: Thank you for your valuable and insightful comment. We have revised those in the manuscript. (Line 13)
- The reviewer doesn’t know why Yin et al and Zhou et al were highlighted in citation of works in Introduction. Surely, there must be a reason for scientific significance or work mostly related to the current research project. No doubt, it’s fine if their works were first reported, not reviewing other’s works. Just making sure. Also, in updating previous discoveries of GRAS family transcriptional factors, more considerations should be focused on the original works, even from reviews.
Author’s response: Thank you for your valuable and insightful comment. We have revised n the manuscript. (Line 37-48)
Furthermore, all changes to the revised manuscript are indicated in the manuscript using track changes. Thank you so much.
Reviewer 4 Report
Comments and Suggestions for Authors
Review IJMS 38775083
The GRAS family is a plant-specific transcription factor family that regulate a myriad of biological functions. In the recent yours this gene family has been elucidated in many plant species. Zheng and colleagues describe the results of a genome wide identification of the GRAS family in Dimocarpus longan, a tropical tree species that produces edible fruits and is distributed throughout tropical and subtropical regions. Using bioinformatic analysis, the authors identified 47 DlGRAS genes in the longan genome distributed on 12 chromosomes which were classified into 10 subfamilies. They presented data on phylogenetic relationships and collinearity, as well as gene structure and cis-elements in DlGRAS promoters in longan.
Through the analysis of different transcriptome datasets, the potential biological functions of these genes in longan somatic embryogenesis (SE) were elucidated, given that SE system offers a viable solution to the challenges associated with longan sampling. The authors performed subcellular localization and promoter functional analysis of two members of the family, DlGRAS10 and DlGRAS22. To investigate the regulatory functions of DlGRAS10/22, Zheng and colleagues also constructed DlGRAS10/22-OE lines and analyzed the expression profiles of several genes involved in hormone synthesis and signaling and simultaneously related to SE, in addition to the ABA and GA3 content in these lines. Results of the study suggest that DlGRAS may be involved in the process of longan SE through hormone-responsive mechanisms
The study provides original and detailed information about the DlGRAS gene family in longan, and contributes to their functional characterization. However, the manuscript needs editing before it accepted for publication in IJMS.
In particular, the authors state: “Therefore, DlGRAS10/22 may be involved in the ABA and GA3 biosynthesis pathways (Line 527), taking into account the expression of DlGID1, DlGA20ox2, DlFUS3, DlABI3 and DlLEC2 Furthermore. “The ABI3 gene can regulate ABA synthesis (Line 600). However, DlABI3 is involved in ABA signaling, while DlGID1 is GA3 receptor.
The statement “During SE, the GRAS gene interacts with other transcription factors, hormone signals and environmental factors to modulate embryonic cell differentiation, proliferation and development “(lines 592-593) is also not accurate, since hormone signals and environmental factors regulate genes rather than interacting with them directly as transcription factors
And finally, “C Determination of IAA and GA3” (line 532) should be substituted for “C Determination of ABA and GA3”.
Author Response
Dear Reviewer:
Thank you very much for your comments. Those comments are all insightful and very helpful for revising and improving our manuscript, as well as the important guiding significance to our research. We have studied comments carefully and have revised the manuscript thoroughly, and the point-by-point responses to the comments are as follows.
All changes to the revised manuscript are indicated in the manuscript using track changes.
- In particular, the authors state: “Therefore, DlGRAS10/22 may be involved in the ABA and GA3 biosynthesis pathways (Line 527), taking into account the expression of DlGID1, DlGA20ox2, DlFUS3, DlABI3 and DlLEC2 Furthermore. “The ABI3 gene can regulate ABA synthesis (Line 600). However, DlABI3 is involved in ABA signaling, while DlGID1 is GA3 receptor.
Author’s response: Thank you so much for your comments. We have revised in the manuscript. (Line 349-351)
- The statement “During SE, the GRAS gene interacts with other transcription factors, hormone signals and environmental factors to modulate embryonic cell differentiation, proliferation and development “(lines 592-593) is also not accurate, since hormone signals and environmental factors regulate genes rather than interacting with them directly as transcription factors
Author’s response: Thank you for your comments. We have rewrite this sentence in the revised manuscript. (Line 411-413)
- And finally, “C Determination of IAA and GA3” (line 532) should be substituted for “C Determination of ABA and GA3”.
Author’s response: Thank you so much for your careful check. We have replaced this sentence in the revised manuscript. (Line 355)
Furthermore, all changes to the revised manuscript are indicated in the manuscript using track changes. Thank you so much.
Round 2
Reviewer 1 Report
Comments and Suggestions for Authors
The authors have addressed all my comments well.
Author Response
Dear Reviewer:
Thank you for your comments, which have significantly raised the quality of the manuscript and have enabled us to promote the manuscript.
If you have any other suggestions for improving our manuscript, please do not hesitate to tell us, and we will rewrite or improve the manuscript by your suggestions. Thank you for your professional review.
Reviewer 2 Report
Comments and Suggestions for Authors
The revised version made targeted adjustments in structure optimization and content supplementation. The overall revision direction conformed to the requirements of academic paper standards. There are still some problems need address.
- “DlGRAS6, DlGRAS10, DlGRAS23 and DlGRAS26 promoted their expression after reatment with 3~12mg/L ABA” should be “The relative expression levels of DlGRAS6, DlGRAS10, upregulated after treatment with 3~12mg/L ABA”. reatment should be treatment? The authors should check the whole manuscript to avoid language mistakes.
- There are still inconsistent use of terminology. For example, "transmembrane domain" is abbreviated as "THM" in Section 2.1 but mistakenly written as "TM" in Section 4.2 (the correct abbreviation is "TM"). "Quantitative real-time PCR" is abbreviated as "qRT-PCR" in the abstract but mistakenly written as "qPCR" in Section 4.7, which does not conform to the academic standard of "writing the full name + abbreviation when first appearing and using the abbreviation uniformly thereafter".
- "To further understand the functional disparities among the promoters of DlGRAS, the PlantCARE online website was used to analyze the cis-elements in the 2000 bp promoter sequence upstream of the initiation codon of the DlGRAS family in longan" (Section 2.4), "online website" is semantically repetitive (PlantCARE itself is an online tool) and can be simplified to "the PlantCARE online tool". In "The results demonstrated that the promoter regions of DlGRAS contained multiple response elements related to plant growth and development, hormone responses, stress responses, and light responses" (Section 2.4), "related to" appears repeatedly and can be optimized to "involved in plant growth and development, hormone responses, stress tolerance, and light signaling".
- In "2.9 Exogenous GA3 and ABA modulate DlGRAS10 and DlGRAS22 promoter activities", only the effects of 50 μM GA3 and ABA on promoter activity are detected, without explaining the basis for selecting this concentration (e.g., whether it is based on the optimal concentration determined in pre-experiments). At the same time, no different concentration gradients (e.g., 10 μM, 50 μM, 100 μM) are set to verify the concentration effect. The reasons for choosing the concentration should be discussed.
- There are still some reference format inconsistent.
- The figure 10 should move to the discussion part.
The authors should check the whole manuscript to improve the language.
Author Response
Dear Reviewer:
Thank you very much for your comments. Those comments are all insightful and very helpful for revising and improving our manuscript, as well as the important guiding significance to our research. We have studied comments carefully and have revised the manuscript thoroughly, and the point-by-point responses to the comments are as follows.
All changes to the revised manuscript are indicated in the manuscript using track changes.
- “DlGRAS6, DlGRAS10, DlGRAS23 and DlGRAS26 promoted their expression after reatment with 3~12mg/L ABA” should be “The relative expression levels of DlGRAS6, DlGRAS10, upregulated after treatment with 3~12mg/L ABA”. reatment should be treatment? The authors should check the whole manuscript to avoid language mistakes.
Author’s response: Thank you so much for your comments. We have revised in the manuscript. (Line 261-262)
- There are still inconsistent use of terminology. For example, "transmembrane domain" is abbreviated as "THM" in Section 2.1 but mistakenly written as "TM" in Section 4.2 (the correct abbreviation is "TM"). "Quantitative real-time PCR" is abbreviated as "qRT-PCR" in the abstract but mistakenly written as "qPCR" in Section 4.7, which does not conform to the academic standard of "writing the full name + abbreviation when first appearing and using the abbreviation uniformly thereafter".
Author’s response: Thank you so much for your careful check. We have carefully reviewed and revised the entire manuscript. Hifair® III 1st Strand cDNA Synthesis SuperMix for qPCR (gDNA digester plus) (Yeasen, Shanghai, China) and HRbioTM qPCR SYBR®Green Master Mix (No Rox) (Heruibio, Guangzhou, China) are two reagents used in the qRT-PCR reaction. (Line 16, 454)
- "To further understand the functional disparities among the promoters of DlGRAS, the PlantCARE online website was used to analyze the cis-elements in the 2000 bp promoter sequence upstream of the initiation codon of the DlGRAS family in longan" (Section 2.4), "online website" is semantically repetitive (PlantCARE itself is an online tool) and can be simplified to "the PlantCARE online tool". In "The results demonstrated that the promoter regions of DlGRAS contained multiple response elements related to plant growth and development, hormone responses, stress responses, and light responses" (Section 2.4), "related to" appears repeatedly and can be optimized to "involved in plant growth and development, hormone responses, stress tolerance, and light signaling".
Author’s response: We gratefully appreciate for your valuable suggestion. We have revised those in the manuscript. (Line 176-180)
- In "2.9 Exogenous GA3 and ABA modulate DlGRAS10 and DlGRAS22 promoter activities", only the effects of 50 μM GA3 and ABA on promoter activity are detected, without explaining the basis for selecting this concentration (e.g., whether it is based on the optimal concentration determined in pre-experiments). At the same time, no different concentration gradients (e.g., 10 μM, 50 μM, 100 μM) are set to verify the concentration effect. The reasons for choosing the concentration should be discussed.
Author’s response: Thank you for your valuable and insightful comment. Based on extensive preliminary experiments, 50 μM was identified as the optimal concentration; therefore, the effects of 50 μM GA3 and ABA on promoter activity were detected.
- There are still some reference format inconsistent.
Author’s response: Thank you so much for your careful check. We have revised this in the revised manuscript. (Line 646, 720, 765)
6.The figure 10 should move to the discussion part.
Author’s response: We gratefully appreciate for your valuable suggestion. We have revised in the manuscript.
Furthermore, all changes to the revised manuscript are indicated in the manuscript using track changes. Thank you so much.